# Critical review of healthcare financing and a survey of system quality perception among healthcare users in Nigeria (2010–2023)

Blessing Osagumwendia Josiah[1,2*¤a], Emmanuel Chukwunwike Enebeli[1,3],
Brontie Albertha Duncan[4], Prisca Olabisi Adejumo[5], Chinelo Cleopatra Josiah[6],
Lordsfavour Anukam[2], Muhammad Baqir Shittu[7], France Ncube[8],
Kelechi Eric Alimele[9], Mercy Emmanuel[10], Oyinye Prosper Martins-Ifeanyi[11],
Fawole Israel Opeyemi[1,12], Oluwadamilare Akingbade[1,¤b], Abosede Peace Adebayo[1],
Busiroh Mobolape Ibraheem[1], Ubiebo Ataisi Ekenekot[13], Mudiaga Sidney Edafiejire[14],
Solomon Oluwaseun Olukoya[15], Ufuomaoghene Jemima Mukoro[16],
Siyouneh Baghdasarian[2], Joy Chioma Obialor[17], Gloria Oluwakorede Alao[18],
Blessing Onyinye Obialor[17], Ndidi Louis Otoboyor[19], Oghosa Gabriel Josiah[20],
Joshua Okonkwo[21], Precious Ebinehita Imoyera[22], Ajao Adewale Gbolabo[23],
Blessing Chiamaka Nganwuchu[24], Olukayode Joseph Oladimeji[25],
Timothy Wale Olaosebikan[26], Marios Kantaris[27]

**1** Institute of Nursing Research, Oshogbo, Nigeria, **2** IUHS School of Medicine, Basseterre, Saint Kitts and Nevis, **3** Leeds Teaching Hospitals NHS Trust, Leeds, United Kingdom, **4** Freelance Financial Consultant, Basseterre, Saint Kitts and Nevis, **5** Faculty of Nursing, University of Ibadan, Oyo State, Nigeria, **6** Windsor University School of Medicine, Canyon, St Kitts and Nevis, **7** Department of Nursing Science, University of Ilorin, Kwara State, Nigeria, **8** UNICAF University, Harare, Zimbabwe, **9** Queen Margaret university, Belfast, Northern Ireland, United Kingdom, **10** Delta State College of Nursing Science Agbor, Delta State, Nigeria, **11** University of Benin, Ugbowo-Lagos Road, Uselu, Edo State, Nigeria, **12** Leadcity University, Ibadan, Oyo State, Nigeria, **13** Rivers State School of Nursing, Port Harcourt, Nigeria, **14** Delta State University Teaching Hospital, Oghara, Nigeria, **15** Ahmadu Bello University, Zaria, Kaduna State, **16** Anchor University, Lagos State, Nigeria, **17** Department of Nursing Science, University of Jos, Plateau State, Nigeria, **18** Ladoke Akintola University of Technology, Ogbomoso, Oyo State, Nigeria, **19** Scripps Hospital San Diego, California, United States of America, **20** University of Sunderland, Sunderland, United Kingdom, **21** Marareeda Services, Jos, Plateau state, Nigeria, **22** Irrua Specialist Teaching Hospital, Irua, Edo State, Nigeria, **23** National Open University of Nigeria, Lagos State, Nigeria, **24** Abia State University Uturu, Abia State, Nigeria, **25** Epitome Megacon Consulting Limited, Redemption City of God, Ogun State, Nigeria, **26** Joseph Ayo Babalola University, Ikeji-Arakeji, Osun State, Nigeria, **27** Health Services and Social Policy Research Centre, Cyprus

¤a Current address: Department of Nursing, Turks and Caicos Islands Community College, Grand Turk, Turks and Caicos Islands.
¤b Current address: Faculty of Nursing, University of Alberta, Canada.
* josiahblessing141@gmail.com

## Abstract

Nigeria aims to enhance its healthcare quality index score of 84th out of 110 countries and its Sustainable Development Goals Index ranking of 146th out of 166. Due to increased population, disease burden, and patient awareness, healthcare demand is rising, putting pressure on funding and quality assurance. The Nigerian healthcare financing and its impacts are complex; this study gives insights into the trends. This questionnaire-based cross-sectional survey (conducted from June to August 2023) and 2010–2023 health budget analysis examined healthcare

**Data availability statement:** In compliance with the applicable reporting standards, the datasets supporting the conclusions of this article are available in the Dryad Digital Repository at http://datadryad.org/stash/share/gZJC4g6cgMgi9cJhwLZaImdEPQGrdh72L7Y-uOWM7pps.

**Funding:** The authors have declared that no competing interests exist.

**Competing interests:** The authors have declared that no competing interests exist.

finance patterns and user attitudes (utilisation, preference and quality perceptions) in Nigeria. Data from government health budgets and a stratified random sample of 2,212 from nine states, obtained from the socioculturally diverse 237 million population, were analysed with a focus on trends, proportions, frequency distributions, and tests of association. Results show that the average rating of healthcare experiences did not vary significantly over the last decade. Healthcare system quality was rated mainly poor or very poor; structure (74.09%), services (61.66%), and cost (60.89%). While 87.36% used government healthcare facilities, 85.00% paid out-of-pocket, and 72.60% of them were dissatisfied with the value for money. Despite a preference for government facilities (71.43%), respondents cited high costs (62.75%), poor funding (85.65%), inadequate staffing (90.73%), and lack of essential medicines (88.47%) as major challenges. The budget analysis reveals an average government healthcare fund allocation of $7.12 compared with an estimated expenditure of $82.75 per person annually. Nigeria allocates only an average of 0.37% of GDP and 4.61% of the national budget to healthcare, comprising a maximum of 13.56% of total health expenditure. This study emphasises the urgent need for policy reforms and implementations to improve Nigeria's healthcare financing and service quality. Targeted interventions are essential to address systemic challenges and meet population needs while aligning with international health services and best standards.

## Introduction

### Background

The need for better healthcare has continuously grown over the past few decades due to increasing patient needs and awareness, rising non-communicable diseases, and the increased cross-border transmission of communicable diseases [1–3]. The ability to provide quality, accessible, and affordable healthcare has become a priority for many countries as they strive to bring the best services to the population by investing in areas that directly or indirectly affect health and healthcare [4,5]. The countries of Taiwan, South Korea, and Australia have been able to meet most of the needs of efficient healthcare in recent years, to emerge as the best three in 2024 according to the CEOWORLD Magazine Health Care Index [4]. Other countries have increased their healthcare capacities, obtaining better ranks over the years. Interestingly, only a few African countries emerged in the top 110 countries, with Tunisia being the highest at 49th, while Nigeria ranked 84th. Similarly, evidence from the Sustainable Development Goals (SDGs) of the United Nations suggests that although Nigeria has made efforts in improving the healthcare systems, the progress towards achieving the SDGs is slow, as the 2024 SDG index placed Nigeria in the 146th position out of 166 countries [6,7]. This article reviews how Nigeria has financed healthcare over the last decade and investigates the users' opinions concerning the quality of care they received.

Historical data and experience show that Nigerian healthcare has not occupied the better positions on the spectrum in performance ratings by healthcare workers and users [8]. A study by Etukumuna and Orie [9] reported that healthcare workers in a Nigerian Teaching Hospital were not satisfied with their working conditions, compensation, and career development. Similarly, another study conducted in Nigeria revealed that more than half of the healthcare professionals in Nigeria reported having a poor quality of life, which was influenced by both personal and work-related reasons [10,11].

While the government has made efforts and maintains that the budgets and allocations to healthcare are being used efficiently to enhance the sector [12], many factors, such as population growth, continue to intensify the complexity of health needs and the requirements to meet them [13]. From the providers' perspective, a mix of low remuneration, inefficient infrastructure, insufficient workforce, and poor administration has been regularly cited by healthcare workers like physicians, nurses, pharmacists, and laboratory scientists [11,14]. Consequently, the healthcare system in Nigeria has experienced a huge exodus of key service providers, leading to massive outbound medical tourism by the upper class, and poor access to quality care by the lower class [15,16]. As Nduka Obaigbena [17] said, "*a healthy nation is a wealthy nation*"; therefore, there is a need to focus attention on the healthcare sector.

Evidence suggests that healthcare financing is one of the major challenges of the Nigerian healthcare system, characterised by low public spending and one of the highest out-of-pocket financing in the world [18]. Effective interventions by the Nigerian Government and the Ministry of Health should be reflected in end-users' reports of the improvement and impacts during the past decade. Analyses of the Nigerian healthcare financing should provide insights into the healthcare quality, as healthcare financing has been associated with the quality of healthcare systems [19]. This study seeks to understand how Nigeria has budgeted annually for healthcare, reflected per 100,000 persons, in the last decade and what public opinion says about the trends in quality of care received in the same period, irrespective of external rankings.

## Review of the Nigerian healthcare system

**General ranking.** From the past two decades of the World Health Organisation (WHO) global health system ranking, Nigeria has done relatively well by moving from 187 out of 190 countries in 2000–149 out of 166 countries in the 2024 analysis [6]. Many attributes of a country's healthcare system are used for the ranking [20–22]. These usually include infrastructure and technology, staffing, health equity and access, administrative efficiency, and key indicators such as population health, mortality, and other quantitative health outcome indices. These indices constitute the quality of a healthcare system from the provider's point of view, which may differ from that of healthcare workers and users. Following national healthcare investments, the key players and major providers may have different expectations about the care outcome, relative to that of the care receivers [23].

**Key performance indices.** Nigeria has been reporting an improving performance in the global space. According to the United States Agency for International Development (USAID) in 2017, the fertility rate in Nigeria was 5.5, and the population growth was 3.2%, being one of the fastest-growing population globally [24]. In 2024, the rates reduced to 4.3 and 2.1, respectively. This reflects the possible impacts of ongoing social transformations and family planning mechanisms in the country and region at large [25,26].

Infant mortality in 2010 was 8.65% (86.46/1000 live births), the third highest globally. In 2024, estimates of 5.26% (52.61/1,000 live births) and 6.77% (67.73/1,000 live births) were reported. Although there is a significant improvement in the statistics, Nigeria has fallen to the second-highest infant mortality rate, only second to Sierra Leone [27,28]. Similarly, in 2016, the maternal mortality was the fourth highest at 576 deaths per 100,000 live births [29]. According to the latest available WHO data, Nigeria's maternal mortality rate ranking became the second worst globally in 2020, with 1,047 deaths per 100,000 live births [30]. Average life expectancy in Nigeria has been rising steadily since 1960 and is currently estimated at between 54.78 and 56.36 years [31–33]. Also, the crude death rate in Nigeria has moved from 14.48 in 2010 to a current estimate of between 10.67 and 11.56 per 1,000 people in 2025, but is still better than that of twenty-four other countries [34–36].

**Healthcare costs.** Besides the numerical ratings, budgeting, expenditures, and process management, like infrastructure development, training, staffing, and insurance systems, are specific areas for ensuring a higher quality of care [37]. These directly impact how much users have to pay for care and the type of care they receive. Health expenditure is one of the leading areas of distress for average Nigerians. According to the Institute for Health Metrics and Evaluation [IHME] [38], the average health spending per person in 2021 was US$83.00, out of which US$62.18 was the out-of-pocket expenditure of the users [39,40]. Although a positive view in the reports, this amount is exorbitant when compared with the US$10.90 covered by the government, the mid-year minimum wage of US$74.80, and the annual GDP per capita of US$2,021 for the same year [41]. This data estimates the average health expenditure for that year as 3.08% per capita GDP [38,41], requiring further investigation.

**Health insurance.** Administrative transparency and governance in finance and expenditure associated with healthcare are powerful ways to positively influence the user experience [42]. For instance, the International Trade Administration [ITA] in 2023 noted that Nigeria's national health insurance service is still considered ineffective, with less than 5% total population coverage. Other measures, such as Health Maintenance Organisations [HMOs], also have challenges with market shares, which necessitate reducing premium prices and consequently the quality of services [43]. As of 2024, Nigeria still celebrates a 19.2 million enrolment on the National Health Insurance Scheme (NHIS), which was 95% of the 2027 presidential target. However, this is only 8.36% of full national coverage [44].

**Products and services.** Healthcare process management and infrastructure delivery have not been relatively different from the pace of general national development and the economic experience in Nigeria. The ITA reported that the Nigerian health infrastructure is still underdeveloped, with no modern medical facilities [45]. It further states that from 2018 to 2022, Nigeria did not produce exportable healthcare-related products or services, but it imported 61 products in 2018, 67 in 2019, 74 in 2020, 84 in 2021, and 94 in 2022 [45].

**Leadership and human resource development.** Nigeria experiences over US$400 million in loss to outbound medical tourism annually, largely due to poor infrastructure and gross understaffing. With a population of over 236 million in 2023, Nigeria had only 1.83 skilled healthcare workers per 1,000 persons, as against the WHO benchmark of 4.45 per 1,000 persons [46,47]. An emerging concern is the quality of training. The effects of inefficient administrative systems adversely impact healthcare and medical education. This leads to the preponderance of paralytic health financing, corrupt political and managerial practices, dead infrastructure, poor remuneration for healthcare workers, inadequate social services, heightened security threats, which relegate medical professionals to a position of low self-confidence in their training, professional skills, and abilities [48]. These factors have left Nigeria's healthcare system in a pitiable state [14,46,49].

**Global standard of practice.** According to Tulchinsky and Varavikova [50], national health systems must fulfil three main functions: "health care delivery, fair treatment to all, and meeting health expectations of the population". Such systems must align with the six WHO-recommended basic building blocks of leadership and governance, healthcare workforce, service delivery, financing and universal access, with a focus on medical products, vaccines, and technologies [50]. Although Nigeria has repeatedly prided itself on improving the global ranking of its healthcare system based on the above parameters, the reality is still daunting, and it is the user experience that has become an important and rapidly emerging key performance indicator of healthcare quality and effectiveness. The goal of national healthcare services should be to ensure that the clients are satisfied with the services and products being supplied [19], as people are becoming more interested in having their voices heard, and not just having policies, rules, regulations, and reviews generated on their behalf [51]. This study will attempt to fill this gap by offering insights into quality perceptions of healthcare users in Nigeria.

### Research purpose

Data from the study will be used to assess how government budgeting has changed over the last decade relative to population dynamics, how people feel about healthcare financing, and how they rate the care they have received. The results will assist in process management, policy review, infrastructure improvements, and quality control.

### Research objectives

- To determine trends in Nigerian healthcare budgeting per 100,000 persons in the last decade

- To estimate the dynamics in healthcare quality over the last decade as perceived by users

- To assess how users rate the quality of the healthcare system in Nigeria

- To assess the users' perceived improvements in the Nigerian healthcare system in the last decade.

- To collate users' suggested major challenges in Nigerian healthcare

- To uncover public suggestions for the improvement of the healthcare system in Nigeria

## Methods

### Design

This research combines a retrospective survey with secondary data analysis. It involved interviewing participants about their views on the country's healthcare system and collating key budgetary data for healthcare over the last 13 years.

### Population and scope

The research focuses on the general population in Nigeria, a West African nation (Fig 1) with an estimated 2025 population exceeding 237 million. The survey involved nine randomly selected States, which had a combined population of 72,025,500 as at the time of the study [52].

Respondents were sampled from two randomly selected representative States in each of five geopolitical zones (except one for the South-east, the smallest geopolitical zone), capturing Nigeria's cultural and economic diversity. The North-east zone was excluded due to logistic difficulties. The States are Kwara and Plateau (North-central); Kano and Kaduna (North-west); Lagos and Oyo (South-west); Delta and Rivers (South-south); and Enugu (South-east geopolitical zone) (Fig 1).

### Inclusion and exclusion criteria

Participants included from the States were only Nigerians or foreigners who had resided in the country for at least ten to twelve years at the time of the research. Adults aged 30 years and above were preferentially selected to ensure that the memory checks fall within events that occurred after they reached 18 years. Only Nigerian population records and health-related budgetary data from 2010 to 2023 were included. Adults who met the inclusion criteria but did not consent were excluded.

### Sample and sampling technique

The sample size for each state was calculated using Lagos, which has the largest adult population, as the reference population proportion. The sample for Lagos was initially determined using the Cochrane formula (Z2*p*q/ e2) [54], where Z = 1.96 at a 95% confidence interval, p = 0.5 (incidence rate), q = 0.5 (non–incidence rate) and e = 0.05 (margin of error). The calculation yielded a sample of 384, which was approximated to 390. The population-to-sample ratio derived from Lagos was then applied to other states, resulting in a target total sample size of 1,830 (Table 1).

### Research tools

An analysis of existing records of budgets and gross expenditures using a Microsoft Excel spreadsheet formed one part of the research process, and a questionnaire in print and electronic format was used for the survey. The questionnaire

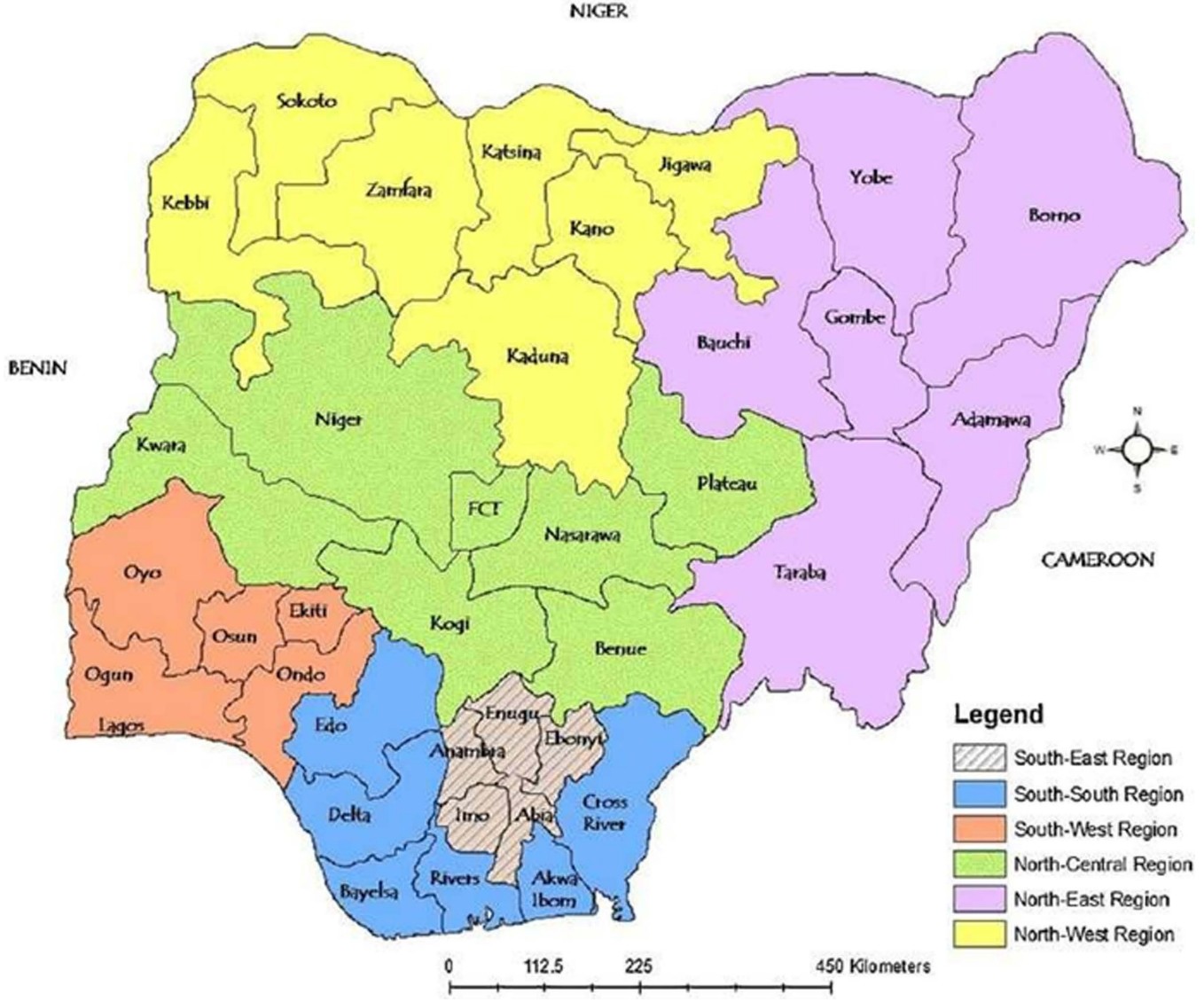

**Fig 1. Nigeria Map: Geopolitical Zones and States (Adapted from Olu-Adeyemi L. Federalism in Nigeria -Problems, Prospects and the Imperative of Restructuring [53]).**

collected sociodemographic data and the views of experiences with healthcare structure, services, and cost, as modified from the Patient Satisfaction Questionnaire (PSQ-18) and the RAND Health Care Patient Satisfaction Survey for the Unified Medical Group Association [55]. A reliability test was conducted using data from a pilot study conducted among 30 adults each in Imo and Edo States, with a Cronbach's alpha of 0.78.

For the secondary data, each domain, such as budgeted funds, expended funds, population sizes, and secondary derivatives, was assigned for inferential analysis, such as a paired T-test between budget and estimated expenditure, a correlational analysis between the total budget and budget per 100,000 people, and a comparative analysis matching budgetary performances against global benchmarks. The sociodemographic attributes were the independent variables, while the users' ratings and facility preferences were dependent variables.

**Table 1. Population and Sample Sizes in Selected States.**

| State (Locations) | Projected General Adult Population | Targeted Sample | Actual Sample |
|---|---|---|---|
| Lagos | 5,295,476 | 390 | 457 |
| Delta | 2,102,587 | 160 | 169 |
| Kwara | 1,091,247 | 90 | 158 |
| Plateau | 1,451,599 | 110 | 114 |
| Kano | 3,999,207 | 310 | 377 |
| Enugu | 1,717,750 | 130 | 135 |
| Kaduna | 2,662,855 | 210 | 260 |
| Rivers | 2,722,049 | 210 | 303 |
| Oyo | 2,885,276 | 220 | 239 |
| Total | 23,928,046 | 1830 | 2212 |

## Data collection

The Federal Ministries of Budget & Economic Planning, Health, and Finance in Abuja were visited for assistance with the budget data. The team was directed to the official digital repository ("*budgetoffice*") for all budget-related information [56,57]. Health finance records were accessed via paid access on *Statista* [58], and from WHO's open health expenditure and financing records [59,60]. Data related to expenditure was calculated using the WHO's *Global Health Expenditure Database*; WHO's *Nigeria: health expenditure as share of GDP*; and the WHO's *Current health expenditure (CHE) as percentage of gross domestic product (GDP) (%) - Data by country* [59,61,62]. This process was carefully executed to ensure accuracy and consistency. It involved the extraction of necessary budget and expenditure data from the respective databases and the application of a multi-step verification and validation process.

The researchers carried out the questionnaire survey with the assistance of trained data collectors from each State between 8/6/2023 and 18/8/2023. Participants were randomly selected from public places using the Spinthewheel app [63] to select between "*select*" and "*exclude*" to ensure randomness of the selected sample, and give all individuals in the sample space a 0.5 chance of being selected. The same approach was applied to social media groups and the blinded lists of respondents from various public spaces. Telephone calls and social media contacts augment the face-to-face processes in the study, especially in some distant areas. Generally, respondents were either interviewed or self-administered the questionnaire after giving informed consent by signing the form provided in print or an electronic version. Data collectors were directly supervised throughout the study to ensure compliance with ethical standards.

## Data analysis

The survey was conducted between 8/6/2023 and 18/8/2023, and the data obtained was collated with Google Forms. The resulting Google sheet was extracted and prepared in Microsoft Excel and SPSS version 29 spreadsheets for analysis. The scale evaluation of the Nigerian healthcare system ratings was assigned as follows: 5 for *Excellent*, 4 for *Good*, 3 for *Average*, 2 for *Poor*, and 1 for *Worse*. Simple frequency distributions and percentages were applied to the sociodemographic data and other variables. Multiple linear regression analysis and the Chi-square test were used to evaluate relationships between the users' perceptions of the healthcare system and several sociodemographic attributes, including the State of residence. The variations within each State were not **particularly** examined due to the unified approach to data collection at the State level.

## Ethics

The study is the budgetary and general population aspect of a compounded proposal submitted as "*Critical review of healthcare financing and end-users' quality perception in Nigeria*", which received approval from the Nigerian Health

Research Ethics Committee (NHREC) with ID: IRB-23–018. The research was designed to independently assess health-care workers' and patients' perspectives on healthcare financing and the quality of care they received using separate approved questionnaires. This ethical process ensured that voluntary participation, anonymity, and secure data storage were observed throughout this research. Written informed consent was obtained from each respondent before they participated in the study, in line with the ethical guidelines in the Declaration of Helsinki and other extant national ethical principles [64].

### Inclusivity in global research

Additional information regarding the ethical, cultural, and scientific considerations specific to inclusivity in global research is included in the Supporting Information (S1 Checklist).

## Results

A total of 2,212 valid responses were collected and analysed with the set objectives of examining trends in Nigerian healthcare budgeting per 100,000 persons in the last decade, users' ratings of the quality of healthcare in Nigeria, improvements, existing challenges, and suggested solutions to the problems in Nigerian healthcare.

Table 2 highlights several key sociodemographic attributes of the respondents. Gender distribution shows a higher percentage of males (57.88%) than females (42.12%). Age-wise, a significant number of respondents fall within the 30–39 years range (39.19%), followed by those aged 40–49 years (36.79%). In terms of state representation, Lagos leads with 20.66%, followed by Kano (17.05%) and Rivers (13.69%). Educational attainment is predominantly at the bachelor's degree level (43.01%), with notable proportions holding Diplomas (19.26%) and Master's/Postgraduate degrees (17.95%).

The employment status reveals that most are full-time (66.05%), while 16.27% are unemployed. The government is the largest employer (37.23%), followed by the private/non-government sectors (26.31%). Income distribution indicates that the highest percentage of respondents (26.44%) earn naira in the range of 50,000 to 99,000, followed by (20.48%) earning 100,000 to 199,000. The largest group works in healthcare (14.51%), education (11.53%) and cultural, legal, and social services (10.08%).

### Utilisation and payment methods for healthcare

Fig 2 shows that the most common frequency of visits is about once every 6–12 months, accounting for 46.11% of respondents. This is followed by visits about once every 1–6 months, representing 30.06% of the respondents. While only 11.17% of respondents use government health facilities more than twice a month, 12.66% do not.

Table 3 shows that only 4.61% of the respondents exclusively used government facilities, and 70.00% combined them with private facilities. In comparison, 12.75% added traditional and non-hospital health interventions, resulting in 87.36% of the respondents patronising the government healthcare facility. A significant majority, 85.00%, reported paying out-of-pocket when using government healthcare facilities, while only 15.00% did not pay out-of-pocket. This indicates a high reliance on personal funds for accessing government healthcare services, which could reflect on the accessibility and affordability of these services. When asked if the services were worth the out-of-pocket payments, only 24.70% of those who paid out-of-pocket felt that the services were worth the cost. In contrast, 72.30% did not think the services justified the payments.

### Preference and perception toward quality of government healthcare systems

Fig 3 illustrates the preferences of healthcare facilities among the respondents. It categorises the types of healthcare facilities by respondents' preference; 71.45% prefer government hospitals, 11.53% prefer private clinics, and 17.04% have no preference (Figs. 4, 5).

**Table 2. Sociodemographic Attributes of Respondents (N = 2212).**

| Attributes | | Frequency | Percentage |
|---|---|---|---|
| **Gender** | Female | 932 | 42.12% |
| | Male | 1280 | 57.88% |
| **Age** | 30-39 years | 867 | 39.19% |
| | 40-49 years | 814 | 36.79% |
| | 50-59 years | 396 | 17.90% |
| | 60 and above | 135 | 6.10% |
| **State** | Delta | 169 | 7.64% |
| | Enugu | 135 | 6.10% |
| | Kaduna | 260 | 11.75% |
| | Kano | 377 | 17.05% |
| | Kwara | 158 | 7.14% |
| | Lagos | 457 | 20.66% |
| | Oyo | 239 | 10.81% |
| | Plateau | 114 | 5.15% |
| | Rivers | 303 | 13.69% |
| **Education** | Bachelor | 951 | 43.01% |
| | Diploma | 426 | 19.26% |
| | High school or below | 360 | 16.27% |
| | Master/Postgraduate | 397 | 17.95% |
| | PhD/Fellowships | 78 | 3.53% |
| **Employment** | Full-time Employment | 1461 | 66.05% |
| | Part-time Employment | 391 | 17.68% |
| | Unemployed | 360 | 16.27% |
| **Employer** | Government | 824 | 37.23% |
| | Private/Not Government | 582 | 26.31% |
| | Self-employed | 432 | 19.53% |
| | Unemployed | 374 | 16.91% |
| **Income** | Less than 35,000 | 358 | 16.28% |
| | 35,000 to 49,000 | 352 | 15.91% |
| | 50,000 to 99,000 | 585 | 26.44% |
| | 100,000 to 199,000 | 453 | 20.48% |
| | 200,000 to 399,000 | 321 | 14.51% |
| | More than 400,000 | 143 | 6.46% |
| **Profession** | Agriculture and Lands | 91 | 4.11% |
| | Banking, Trade, Marketing, and Financial Services | 219 | 9.90% |
| | Cultural, Legal, and Social Services | 223 | 10.08% |
| | Education | 255 | 11.53% |
| | Engineering, Mining, and Architecture | 181 | 8.18% |
| | Entertainment and Arts | 171 | 7.73% |
| | Government, Politics, and Administration | 160 | 7.23% |
| | Healthcare | 325 | 14.69% |
| | Information, Technology, and Communication Services | 188 | 8.50% |
| | Student | 207 | 9.36% |
| | Transportation and logistics | 11 | 0.50% |
| | Unemployed | 181 | 8.18% |

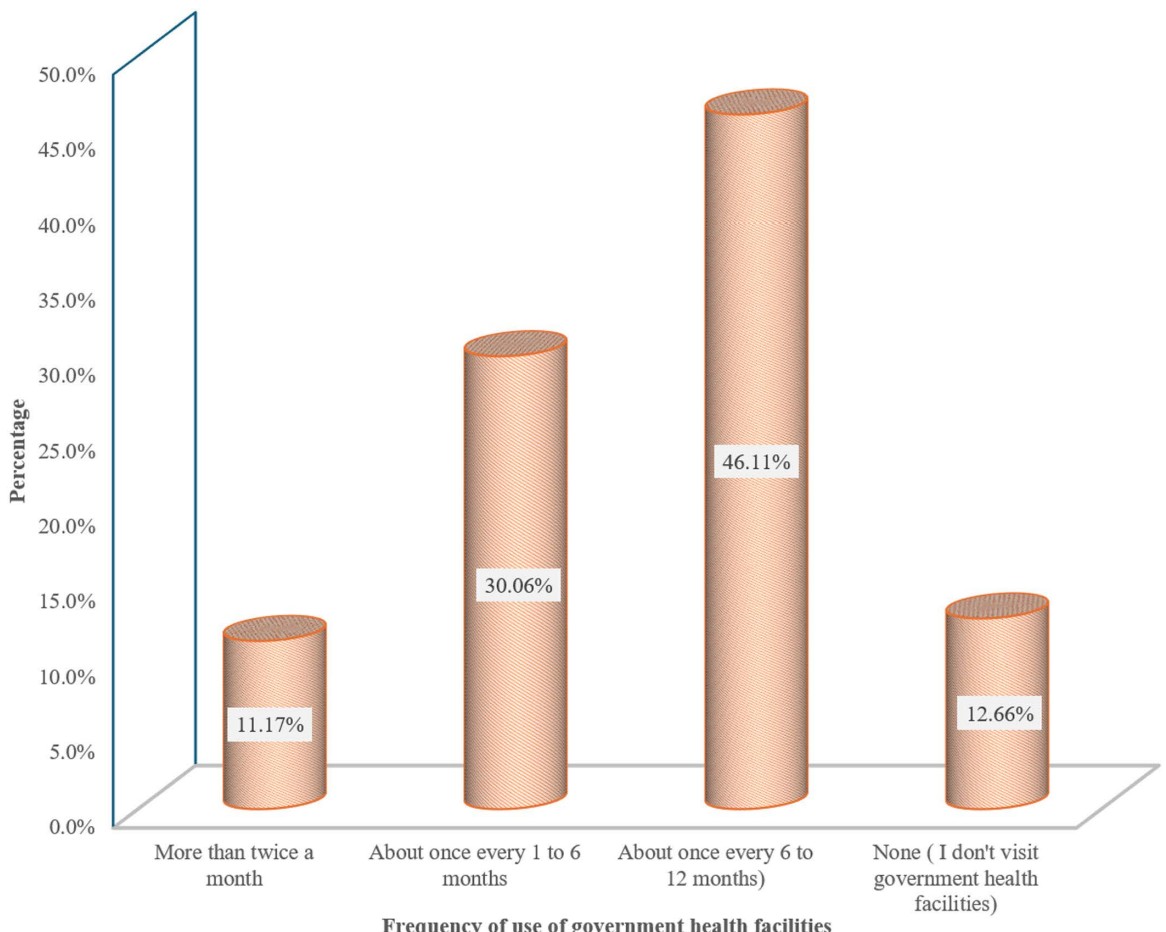

**Fig 2. Frequency of Healthcare Utilisation among Respondents (n = 2212).**

The Nigerian healthcare system faces significant challenges, with most respondents expressing dissatisfaction with its structure, services, and cost. Specifically, 74.09% rated the healthcare structure as poor or worse, while 61.66% rated services similarly. Regarding cost, 60.89% of respondents considered it poor or worse, and 25.36% rated it as average. While there were some positive ratings, indicating that some individuals had satisfactory experiences, the overall picture is one of a system that requires significant improvement to meet the needs of its population.

The graph shows that the quality of Nigerian healthcare systems has fluctuated over the past decade, with no clear upward or downward trend. The " *Poor* " ratings remained relatively high, with an average of around 40.00% and an upward trend leading from the highest share of 49.68% in 2013 to 36.71% in 2023. "*Good*" ratings also maintained a relatively high but upward trend from 33.05% for 2013, peaked at 56.60% for 2022, then dropped to 40.14% for 2023. "*Excellent*" and "*very poor*" did not receive as many votes as the others, with *excellent* moving undulating from 10.67% to 19.48% between 2013 and 2023, while *very poor* moved in the opposite direction from 6.60% to 3.66%.

**Nigeria healthcare budget trends from 2010 to 2023**

Fig 6 presents a compelling analysis of the relationship between Nigeria's Gross Domestic Product (GDP) and Total Health Budgets over the past decade. The data reveals a clear positive correlation, indicating that the government

**Table 3. Utilisation and Payment Methods for Healthcare Services (n = 2212).**

|  |  | Frequency | Percentage |
|---|---|---|---|
| Select the healthcare facility or facilities you or your relatives have used in the past 10–12 years | Government only | 102 | 4.61 |
|  | Government and Private hospitals | 1548 | 70.00 |
|  | Private hospitals only | 216 | 9.77 |
|  | Traditional and non-hospital care, like a prayer house only | 64 | 2.89 |
|  | All the facilities above | 282 | 12.75 |
|  | Total | 2212 | 100 |
| If you or your relatives use government healthcare facilities, do you (or they) pay out-of-pocket? | Yes | 1642 | 85.00% |
|  | No | 290 | 15.00% |
|  | Total | 1932 | 100 |
| If you or your relatives use government healthcare facilities and pay out-of-pocket, do you feel the services were usually worth your payment? | Yes | 450 | 27.40% |
|  | No | 1192 | 72.60% |
|  | Total | 1642 | 100.0 |

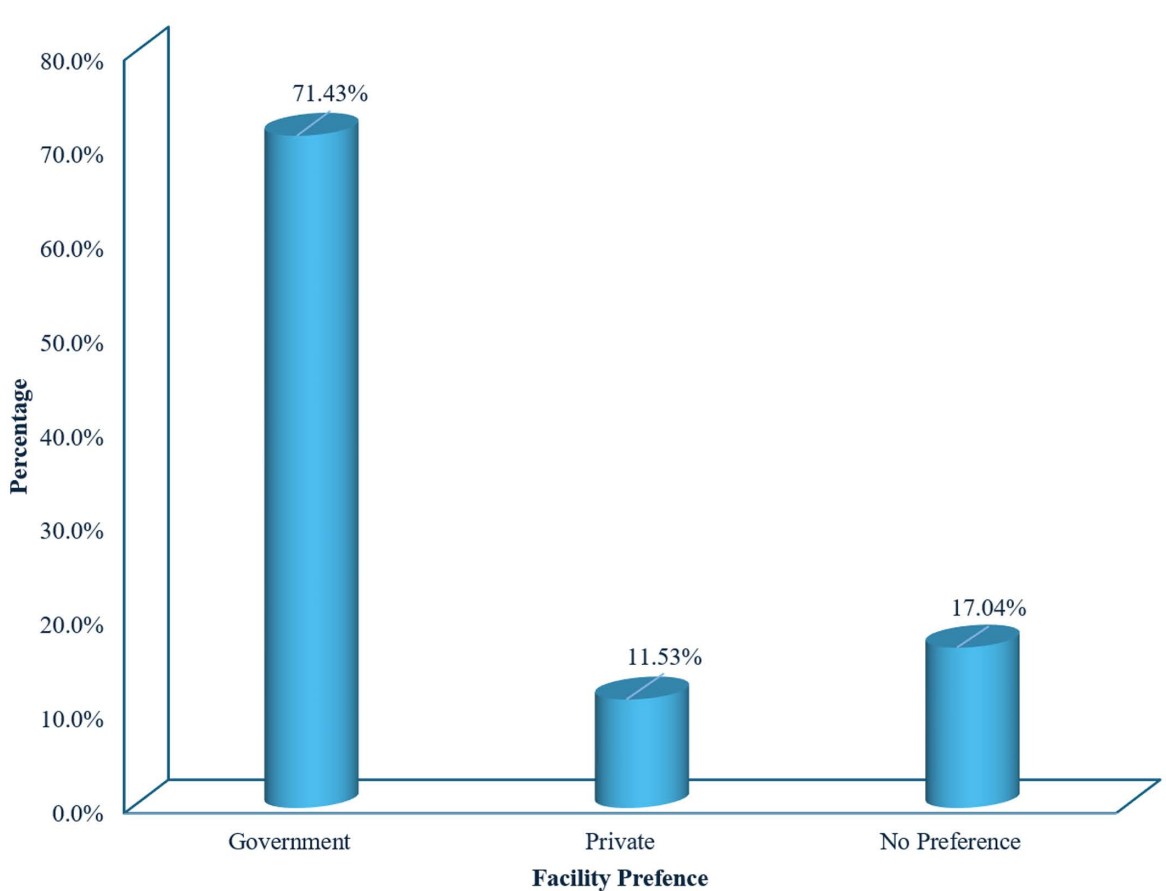

**Fig 3. Preference of Healthcare Facilities among Respondents (n = 2212).**

**Fig 4. Respondents' Rating of the Nigerian Healthcare System (Structure, Services, and Cost) (n = 2212).**

allocated a larger portion of its resources to healthcare as the economy grew. While the GDP experienced a temporary decline in 2013, the overall trend has been upward, and the healthcare budget has continued to increase regardless of these fluctuations (Table 4).

The results of the unpaired T-Test indicate a highly significant statistical difference (t(24) = 8.0312, p < .0001) between the Total Health Budget (THB) and the Estimated Health Expenditures (EHE), with a two-tailed P value of less than 0.0001. Although both variables increased during the period, on average, the EHE exceeded the THB by approximately 3.03 trillion with a 95% confidence interval of approximately 2.25 trillion to 3.81 trillion. There is a substantial and reliable difference between the two variables with t = 8.0312 and a standard error of difference of

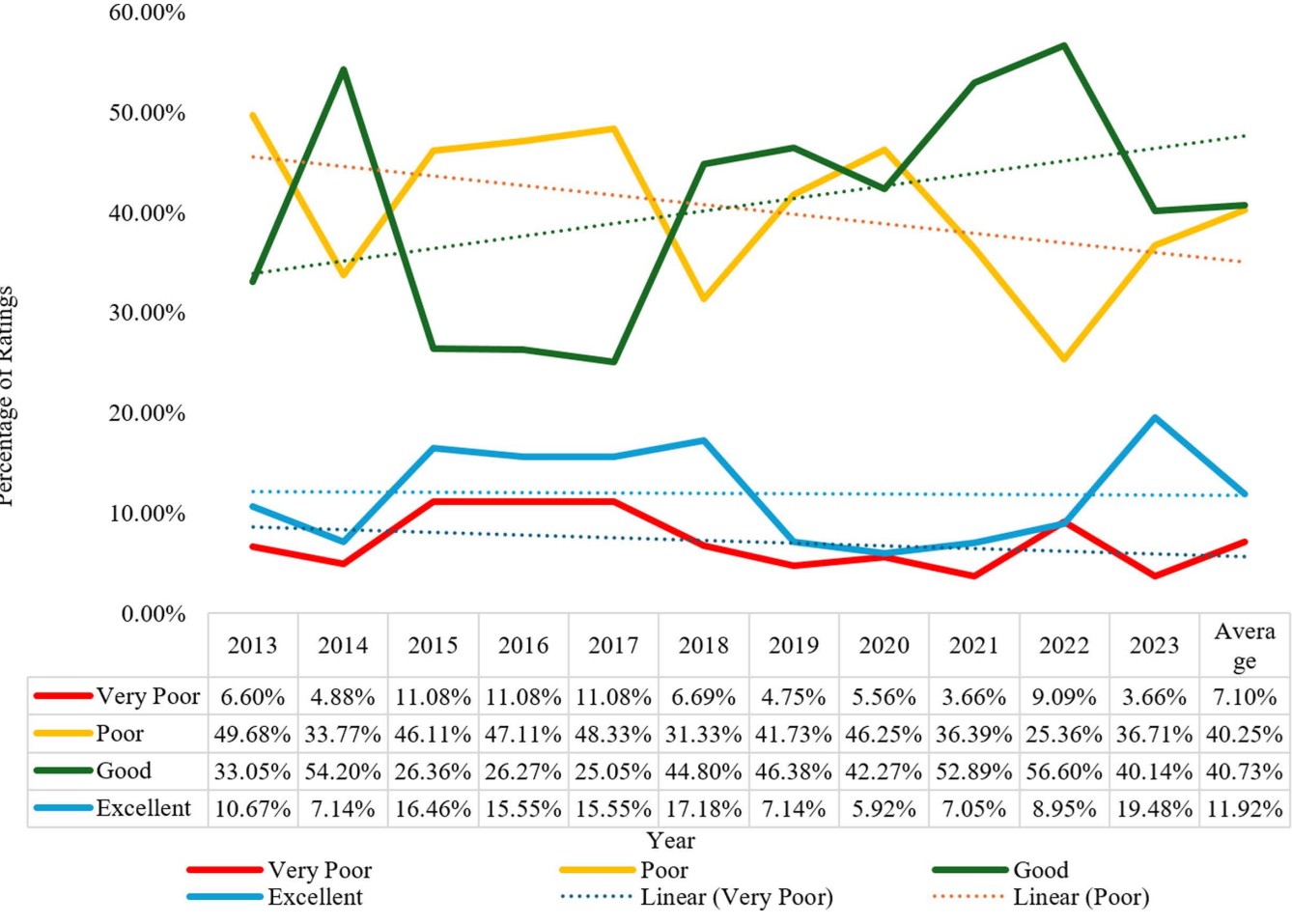

**Fig 5. A summary of respondents' rating of the experience with the quality of Nigerian healthcare systems from 2013 to 2023 (n = 2212).**

377,100,417,332.85, implying that there is an underestimation of the health budget compared to the actual expenditures, highlighting a significant discrepancy that needs to be addressed to ensure accurate financial planning and resource allocation in healthcare.

The data in Fig 8 and Table 5 show a strong positive correlation of 0.997 between the Total Health Budget (mean: 383,246,889,529.25, standard deviation: 228,714,690,393.10) and the per capita health budget (mean: 193,263,675.89, standard deviation: 931,566,79.807) in Nigeria from 2010 to 2023. This time series plot and statistical measures further support this strong relationship, suggesting that increases in the Total Health Budget (N = 14) are closely linked to per capita health budget (N = 14).

Fig 9 compares the Nigerian health budget performance against global benchmarks from 2010 to 2023. The health budget as a percentage of the recommended 5% of GDP fluctuated, peaking at 13.57% in 2011 and declining to 3.46% in 2023. Similarly, the health budget as a percentage of the national budget declined from 5.95% in 2012 to 3.46% in 2022 and then rose to 4.93%. Notably, the health budget has consistently fallen short of the recommended 5% of GDP benchmark, indicating a need for increased investment in the healthcare sector to meet global standards.

Fig 10 shows that Nigeria's estimated total health expenditure as a percentage of GDP increased from 3.47% in 2013 to 4.06% in 2021. While the data fluctuates, especially in the preceding years, like 2012, when it peaked at 5.97% and,

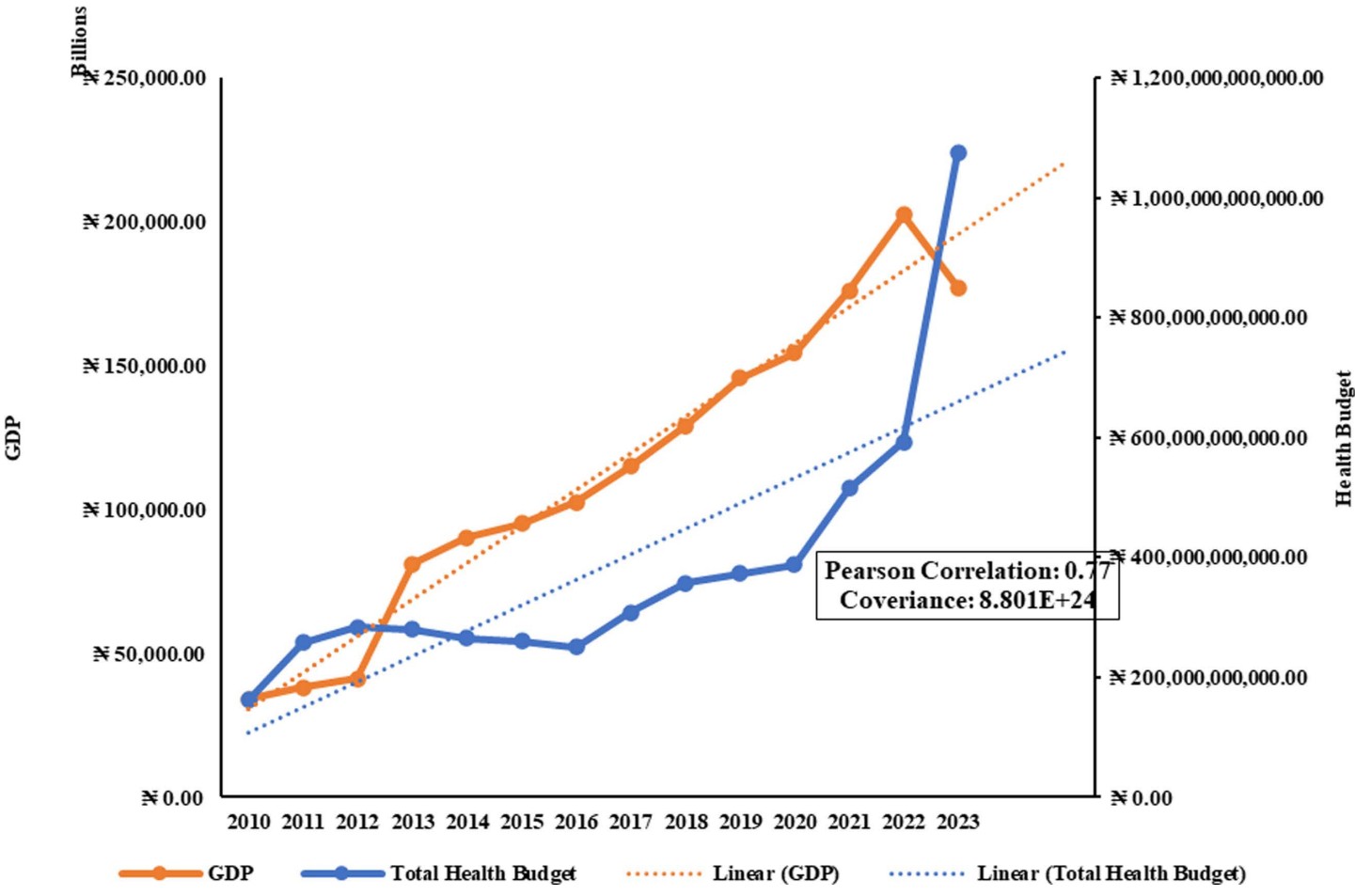

**Fig 6.  A time-series representation of Nigerian Gross Domestic Product (GDP and Total Health Budgets [57,58].**

**Table 4.  Paired T-Test and Correlation between Total Health Budget and the Estimated Annual (Aggregated/Crude) Health Expenditure in Nigeria from 2010 to 2023.**

| Unpaired T-Test | | | | | | | P-Value |
|---|---|---|---|---|---|---|---|
| | **Total Health Budget** | **Estimated Health Expenditures** | **t** | **df** | **95% Confidence Interval of the Difference** | | |
| | | | | | Lower | Upper | |
| Mean | 383,246,895,925.2857 | 3,411,817,555,798.6150 | 8.0312 | 24 | 3028570659873.3293 | 380667663121.2790 | <0.0001 |
| SD | 228,714,690,393.1038 | 1,393,903,473,794.3289 | | | | | |
| SEM | 61,126,572,198.0766 | 402,385,272,909.7551 | | | | | |
| N | 14 | 12 | | | | | |

more recently, at 3.47%, after which it maintained an unsteady rise. It suggests that Nigerians are gradually prioritising healthcare spending regardless of low government input.

The graph (Fig 11) shows a downward trend in the Nigerian annual healthcare budget as a percentage of total health-care expenditure from 2010 to 2021. Although there were spikes in 2011, 2018, and slightly in 2021, the percentage

decreased from 10.82% in 2010 to 7.20% in 2021. This indicates a decreasing commitment to healthcare spending relative to increasing expenditure, as shown in Fig 3.

Table 6 provides a breakdown of the financial outlook of 2212 respondents, showing that a small portion (34.35%) have a sufficient or excess income (8.18%+26.17%), while 65.65% face some varying levels of financial challenges that warrant relying on loans, grants, or family support. Specifically, 9.13% have no sufficient income and no access to loans, 16.09% have negligible income and heavily depend on loans, 40.42% have a reasonable income but sometimes rely on loans.

### Qualities of the Nigerian healthcare system

**Perceived government efforts.** Table 7 provides a breakdown of the perceived government efforts to improve the Nigerian healthcare system, based on the responses of 2212 respondents. The most perceived government efforts are increased staffing (61.88%), built/increased health centres (46.38%), and built/increased hospitals (50.68%). Other less identified efforts include providing insurance, increasing tertiary/teaching facilities, providing free healthcare, subsidising healthcare costs, training healthcare providers, and improving their remunerations. While some respondents perceive these efforts, a significant portion also believe they are insufficient or ineffective, suggesting further action is needed to address the challenges.

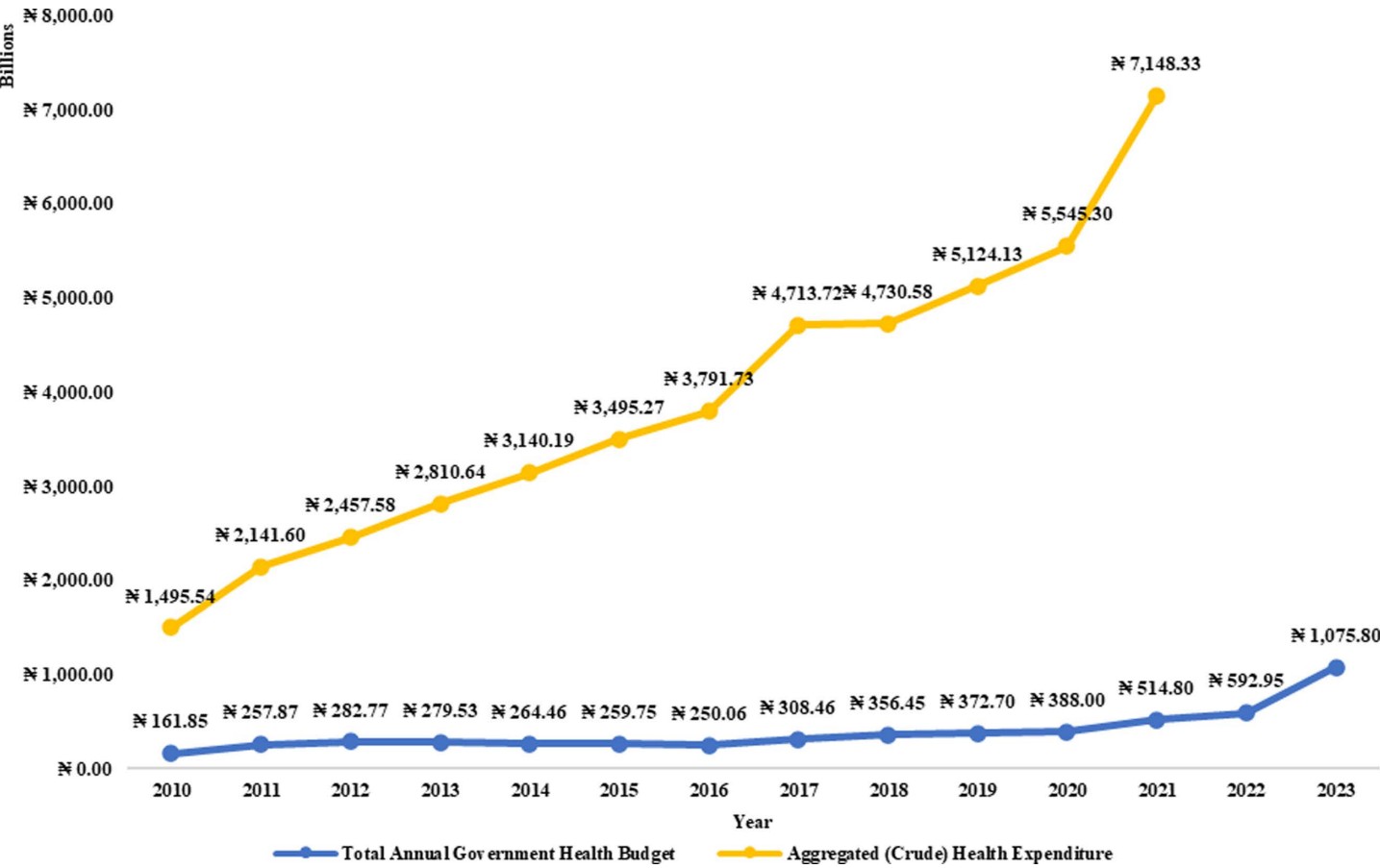

**Fig 7. Time-series representation of Total Annual Health Budget and Aggregated (Crude) Health Expenditure in Nigeria [57,58].**

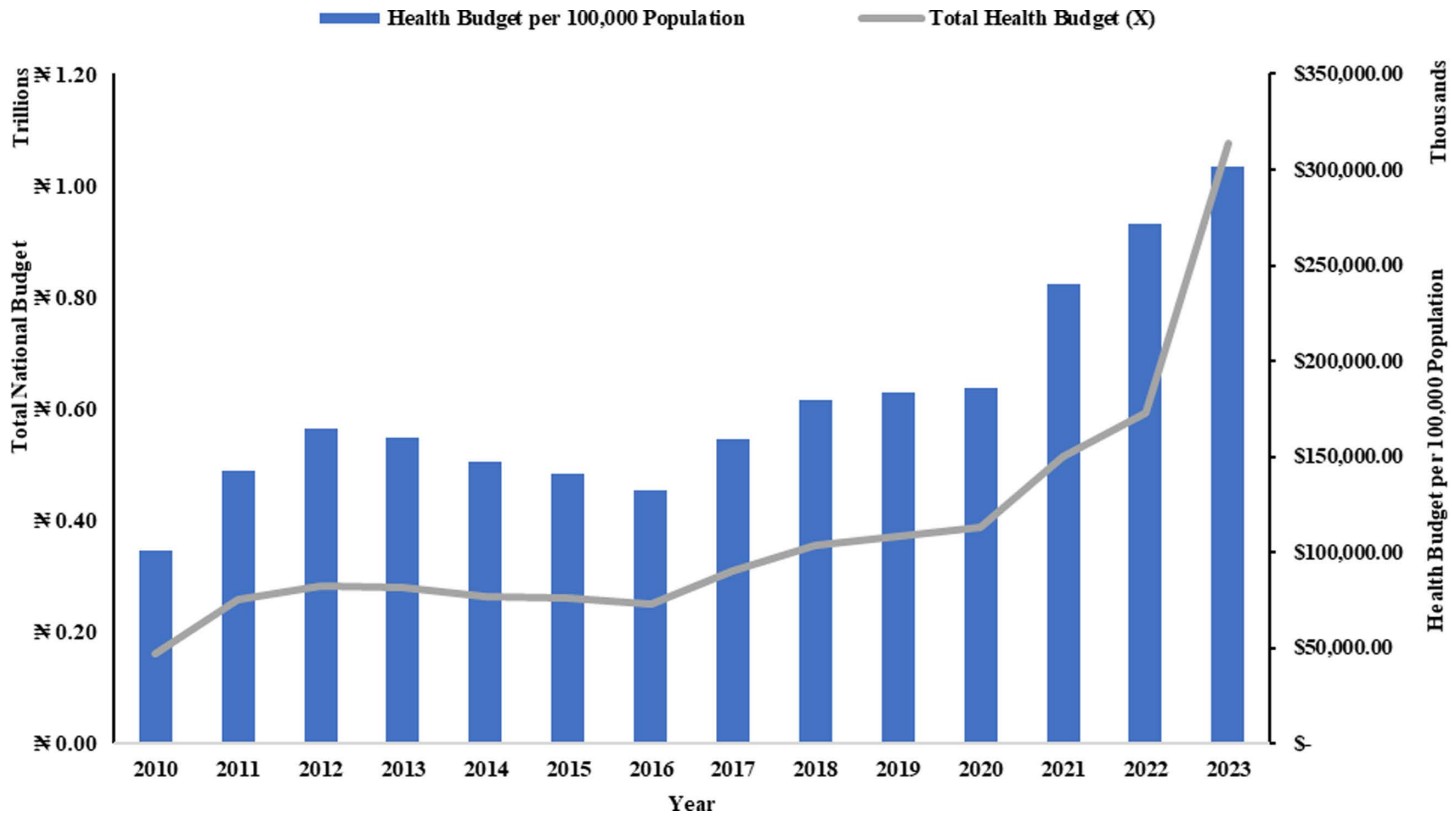

**Fig 8. Representation of Nigerian Health Budget Per 100,000 population and Total Healthcare Budget.**

**Table 5. Correlation analysis between Total Healthcare Budget and the Health Budget per 100,000 Population in Nigeria from 2010 to 2023 [57].**

|  | Mean | Std. Deviation | N | Pearson Correlation | P-Value |
|---|---|---|---|---|---|
| Total Health Budget | 383246895925.29 | 228714690393.104 | 14 | 0.997 | <0.001 |
| Health Budget per 100,000 Population* | 193263765.89 | 93156679.807 | 14 |  |  |

*The per capita budget was achieved through the respective annual population size.*

## Perceived major challenges

Table 8 provides a breakdown of the perceived major challenges in the Nigerian healthcare system, based on the responses of 2212 respondents. The most identified challenges include poor services (92.76%), limited/poor staffing (90.73%), lack of working tools and facilities (91.19%), and corrupt administration and fund misappropriation (92.36%). Other significant challenges identified include high cost, poor funding, incompetent healthcare professionals, long distances to healthcare facilities, substandard pharmaceutical and health products, and lack of essential medicines.

## Suggested further improvement required

Table 9 provides a breakdown of the perceived key areas for improvement in the Nigerian healthcare system, based on the responses of 2212 respondents. An overwhelming majority of respondents agreed that the healthcare system

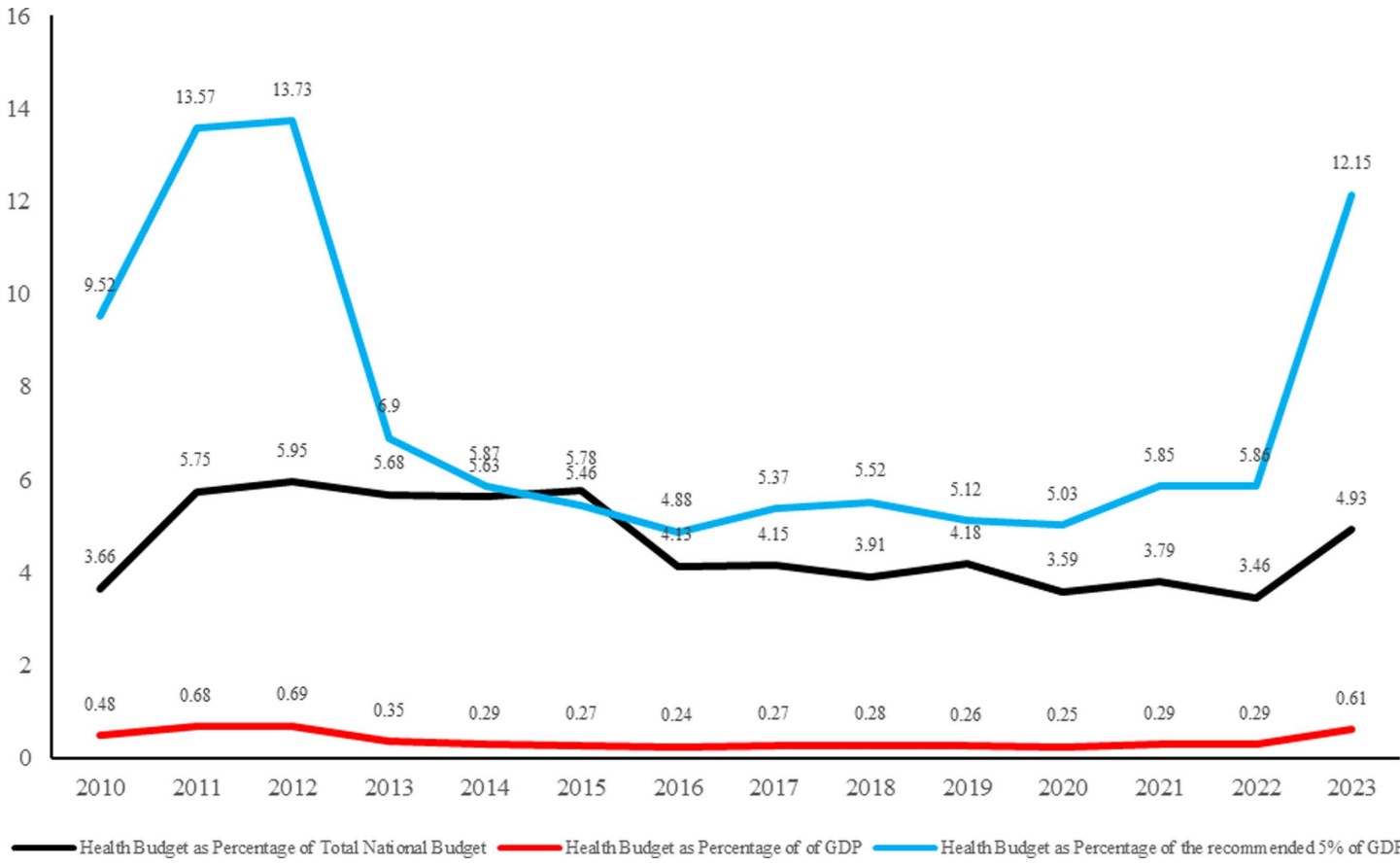

**Fig 9. Comparative Overview of Nigerian Health Budget Performance against Global Benchmarks [57].**

needs to employ more staff (96.12%), build more facilities (96.97%), and increase the budget for healthcare (98.10%). Other significant areas identified for improvement include building a transparent administrative arm (98.00%), improving training, research, and development (97.91%), and increasing the maintenance of facilities and infrastructure (97.41%) (Table 10).

## Factors associated with the users' perceptions

The chi-square analysis reveals that a wide range of sociodemographic factors are significantly associated with the users' ratings of the healthcare system and their preferences for healthcare facilities in Nigeria. These factors for rating are gender ($\chi^2(4) = 14.278$, $p = 0.006$), age group ($\chi^2(12) = 40.783$, $p < 0.001$), state ($\chi^2(32) = 601.459$, $p < 0.001$), education ($\chi^2(16) = 56.536$, $p < 0.001$), employment ($\chi^2(8) = 26.219$, $p < 0.001$), employer ($\chi^2(8) = 27.060$, $p < 0.001$), income ($\chi^2(20) = 121.494$, $p < 0.001$), and profession ($\chi^2(48) = 97.235$, $p < 0.001$). For healthcare preference the significant association are age group ($\chi^2(6) = 40.895$, $p < 0.001$), state ($\chi^2(16) = 353.460$, $p < 0.001$), education ($\chi^2(8) = 31.266$, $p < 0.001$), employment ($\chi^2(4) = 24.690$, $p < 0.001$), employer ($\chi^2(4) = 27.192$, $p < 0.001$), income ($\chi^2(10) = 770.049$, $p < 0.001$), and profession ($\chi^2(24) = 44.442$, $p < 0.007$).

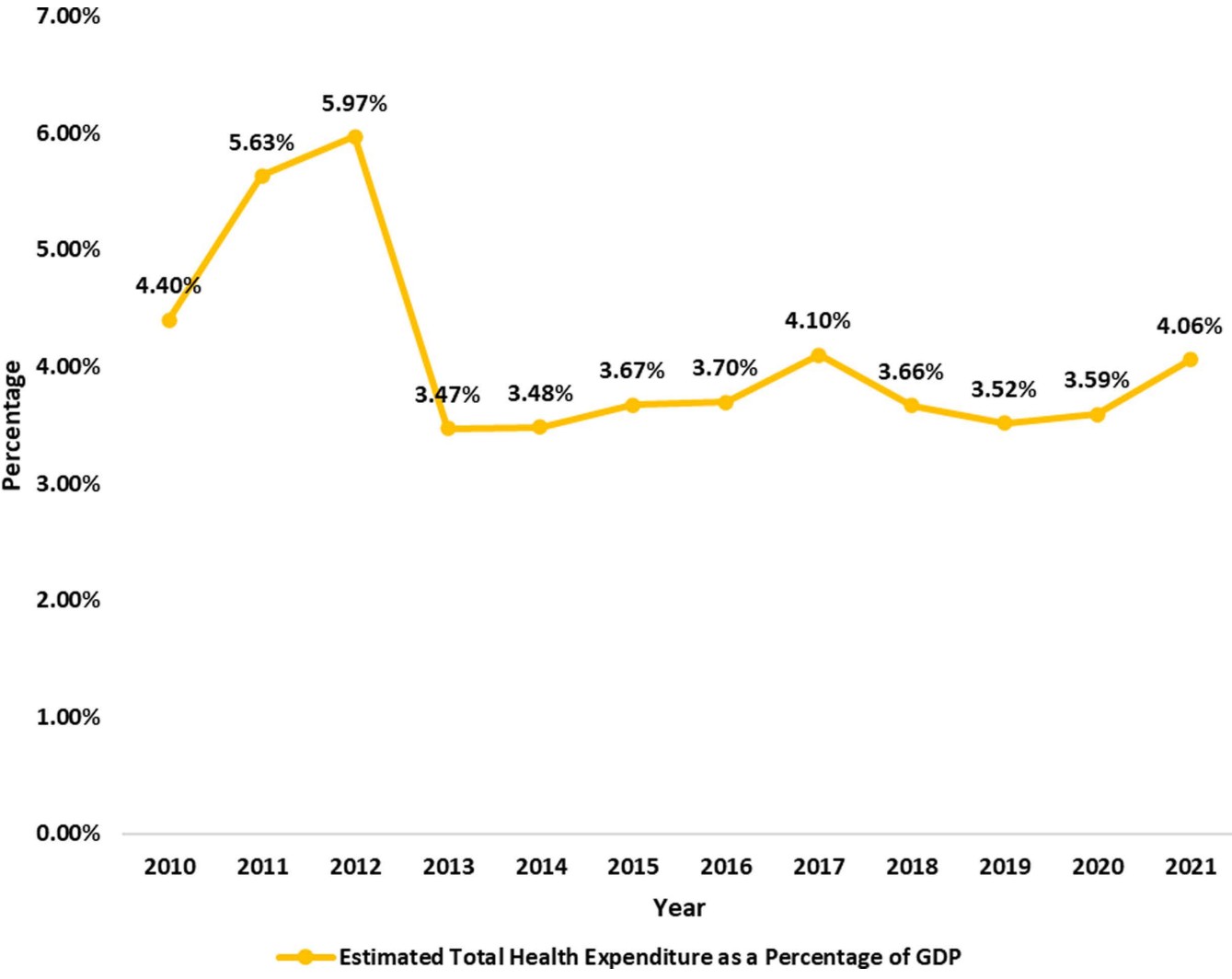

**Fig 10. The Estimated Total Health Expenditure in Nigeria as a Percentage of the Gross Domestic Product (GDP) ([58,65]).**

## Discussion

The findings of this study provide a comprehensive overview of the state of healthcare financing and user perceptions in Nigeria over the past decade. The analysis reveals a persistent disconnect between government healthcare budgets and the realities faced by users, highlighting significant challenges that continue to affect service delivery.

### Utilisation and payment methods for healthcare

Regarding healthcare utilisation and payment methods, the survey reveals significant insights into the patterns of healthcare access among respondents. The data indicates that most individuals utilise government health facilities in conjunction with private healthcare options, with 70.00% of respondents indicating they have used both facilities over the past decade. This combined usage suggests a reliance on government facilities, likely due to their accessibility and affordability, despite only 4.61% exclusively utilising government services.

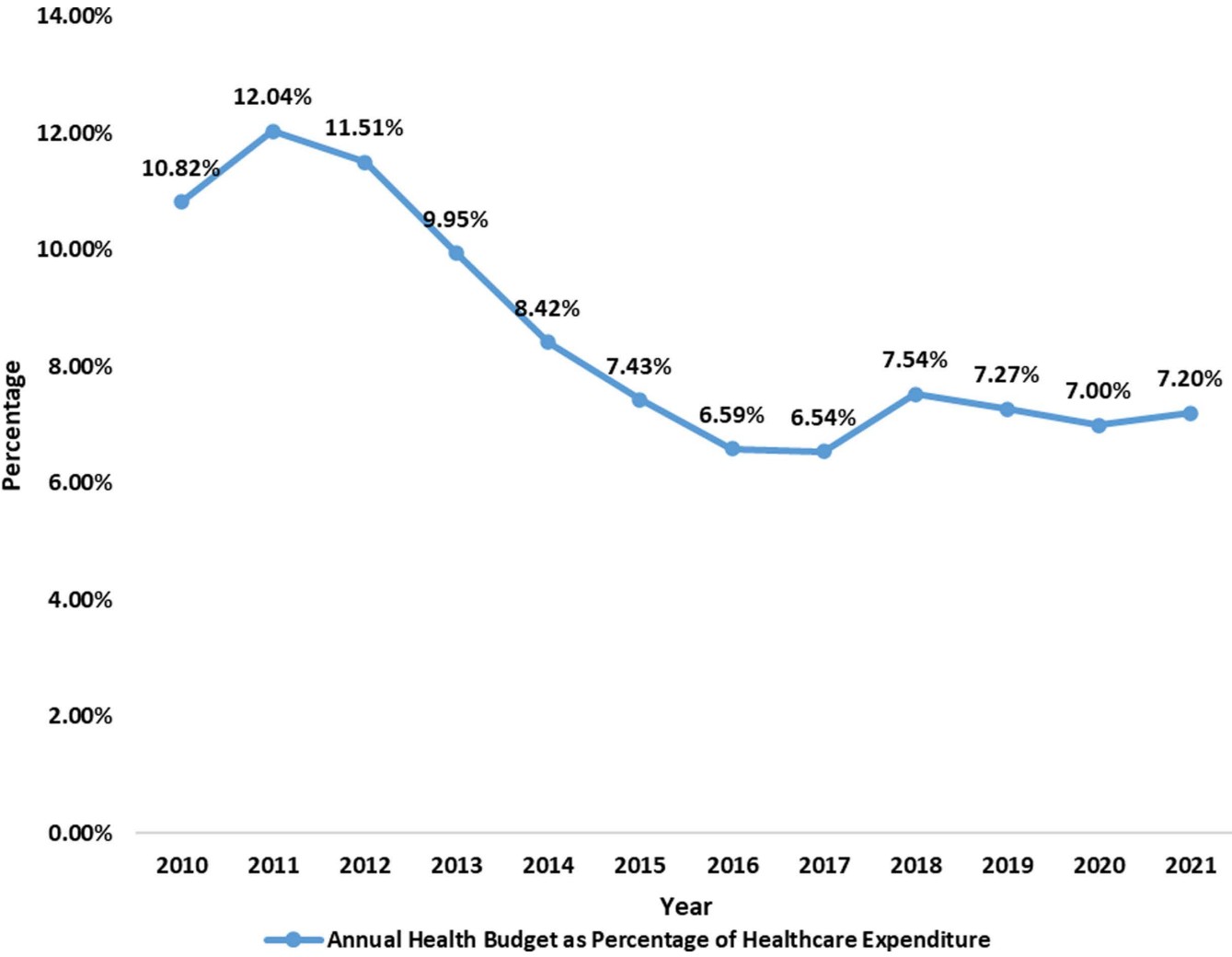

**Fig 11. Nigerian Annual Healthcare Budget as Percentage of the Estimated Total Healthcare Expenditure.**

**Table 6. Financial Attributes and Self-Reported Categorisation of the Respondents (n = 2212).**

| Category | Financial Outlook | Frequency | Percentage |
|---|---|---|---|
| Inadequate | No sufficient income and no access to loans/grants | 202 | 9.13 |
| Minimal | Negligible income and heavy dependence on loans for basic needs | 356 | 16.09 |
| Moderate | Reasonable income, and sometimes dependent on personal and family loans | 894 | 40.42 |
| Adequate | Sufficient income for self and dependents | 579 | 26.17 |
| Surplus | More than sufficient income for self and dependents | 181 | 8.18 |
| **Total** | | **2212** | **100.0** |

**Table 7. Identified Government Efforts towards Improving the Nigerian Healthcare Systems.**

| Healthcare Improvements | Yes | No | Total |
|---|---|---|---|
| Increased staffing | 843 (38.12) | 1369 (61.88) | 2212 |
| Provided insurance | 771 (34.86) | 1441 (65.14) | 2212 |
| Built/increased health centres (PHCs) | 1026 (46.38) | 1186 (53.62) | 2212 |
| Built/increased secondary healthcare facilities (hospitals) | 1122 (50.68) | 1090 (49.32) | 2212 |
| Increased tertiary/teaching facilities (e.g. teaching hospitals, schools) | 994 (44.94) | 1218 (55.06) | 2212 |
| Provided completely free healthcare | 703 (31.78) | 1509 (68.22) | 2212 |
| Subsidised healthcare costs | 1167 (52.75) | 1045 (47.25) | 2212 |
| Prioritised and increased the training of healthcare providers | 939 (42.46) | 1273 (57.54) | 2212 |
| Increased remunerations for healthcare providers | 719 (32.51) | 1493 (67.49) | 2212 |

**Table 8. Identified Prevalent Challenges in the Nigerian Healthcare System.**

| Major Challenges | Yes | No | Total |
|---|---|---|---|
| High cost | 1388 (62.75) | 824 (37.25) | 2212 |
| Poor funding | 1895 (85.65) | 317(14.35) | 2212 |
| Poor services | 2052 (92.76) | 160 (7.24) | 2212 |
| Limited/poor staffing | 2007 (90.73) | 205 (9.27) | 2212 |
| Incompetent healthcare professionals | 1839 (83.13) | 373 (16.87) | 2212 |
| Poor administrative decisions | 1942 (87.80) | 270 (12.20) | 2212 |
| Lack of working tools and facilities | 2017 (91.19) | 195 (8.81) | 2212 |
| Long distance to healthcare facilities | 1639 (74.10) | 573 (25.90) | 2212 |
| Substandard pharmaceutical and health products | 1909 (86.30) | 303 (13.70) | 2212 |
| Corrupt administration and fund misappropriation | 2043 (92.36) | 169 (7.64) | 2212 |
| Lack of essential medicines | 1957 (88.47) | 255 (11.53) | 2212 |

**Table 9. Suggested Key areas for further Improvement in the Nigerian Healthcare Systems.**

| Suggested Solutions | Yes | No | Total |
|---|---|---|---|
| Employ more staff | 2126 (96.12) | 86 (3.89) | 2212 |
| Build more facilities | 2145 (96.97) | 67 (3.03) | 2212 |
| Increase the budget for healthcare | 2170 (98.10) | 42 (1.90) | 2212 |
| Build a transparent administrative arm | 2167 (98.00) | 45 (2.00) | 2212 |
| Increase training, research, and development | 2166 (97.9) | 46 (2.1) | 2212 |
| Increase the maintenance of the facility and infrastructure | 2155 (97.4) | 57 (2.6) | 2212 |

Payment methods further highlight the financial burden associated with healthcare access. A substantial 85.00% of respondents reported paying out-of-pocket for services received at government facilities, reflecting high dependence on personal finances for healthcare. This reliance raises concerns about the affordability and accessibility of these services, particularly considering that only 27.40% felt the services received were worth the out-of-pocket costs. The high prevalence (72.60%) of dissatisfaction with the service's value suggests a disconnect between service quality and patient expectations.

These results align with broader discussions in healthcare literature regarding service utilisation and patient satisfaction. Previous studies have indicated that high out-of-pocket expenses can deter individuals from seeking necessary care,

**Table 10. Chi-square analysis of the sociodemographic predictors of attitude towards the users' rating and preference of healthcare facilities (n = 2212) (See S1 Table and S2 Table for further details).**

| Chi-Square Tests | | | | | | | | | |
|---|---|---|---|---|---|---|---|---|---|
| Factors | | Gender | Age Group | State | Education | Employment | Employer | Income | Profession |
| Average Rating of Healthcare System | P-Value | 14.278 | 40.783 | 601.459 | 56.536 | 26.219 | 27.060 | 121.494 | 97.255 |
| | df | 4 | 12 | 32 | 16 | 8 | 8 | 20 | 48 |
| | P value | 0.006 | <0.001 | <0.001 | <0.001 | <0.001 | <0.001 | <0.001 | <0.001 |
| Preference of Healthcare Facility | Value | 3.428 | 40.895 | 353.460 | 31.266 | 24.690 | 27.192 | 770.049 | 44.442 |
| | df | 2 | 6 | 16 | 8 | 4 | 4 | 10 | 24 |
| | P value | 0.179 | <0.001 | <0.001 | <0.001 | <0.001 | <0.001 | <0.001 | 0.007 |

Significant at 95% interval (<0.05).

potentially leading to worse health outcomes [66,67]. Moreover, dissatisfaction with service value can contribute to lower utilisation rates in the future, as patients may seek alternatives or forgo care altogether [68–72].

The data underscores critical areas for improvement in government healthcare services, including enhancing service quality and addressing cost barriers to ensure that patients perceive value in their healthcare expenditures. Addressing these issues is essential for fostering trust and improving overall community health outcomes.

## User perceptions of healthcare quality

User perceptions of healthcare quality reveal a mixed landscape. While some respondents reported improved access to services, a significant proportion expressed dissatisfaction with overall care quality. Approximately 47.10% rated the healthcare system as "*poor*" or "*very poor*", underscoring widespread discontent with service delivery over the decade. This dissatisfaction is echoed in literature that highlights the gap between provider assessments and patient experiences [23]. Negative user experiences are influenced by factors such as long wait times, inadequate facilities, and perceived neglect from healthcare providers.

Furthermore, studies indicate that user satisfaction is increasingly becoming an important performance indicator of healthcare quality. As noted by Cohen [51], people are becoming more interested in having their voices heard regarding their care experiences rather than relying solely on regulations generated on their behalf. This shift emphasises the need for healthcare systems to prioritise user feedback in their decision-making processes. Additionally, research by Awosoga et al. [10] indicates that healthcare workers report low job satisfaction due to poor working conditions and inadequate compensation. This dissatisfaction among providers can directly impact patient care quality and user perceptions. The interdependence between provider satisfaction and user experiences highlights a critical area for intervention, as earlier highlighted in the healthcare workers component of this study [11].

## Healthcare budgetary analysis

The analysis of Nigeria's healthcare budget trends from 2010 to 2023 reveals critical insights into its commitment to health financing and its implications for public health outcomes. The data presented in Fig 6, Fig 7, Fig 8, Fig 9, Fig 10, and Fig 11 with the accompanying statistical analyses illustrate progress and persistent challenges in healthcare funding.

## Trends and discrepancies in budget allocation and expenditure

Although the weak positive correlation (r = 0.319) suggests that both variables are moving in the same direction, there was a statistically significant difference between the total health budget and estimated annual health expenditure, indicating a

substantial gap in actual spending versus allocated funds. As the budgets gradually increase, the already higher expenditures increase, indicating the need for financial interventions to close the gaps. These discrepancies raise strong concerns about the budgeted funds not adequately supporting healthcare delivery.

Also, despite nominal increases in healthcare budgets, the per capita health budget faced a downward turn from 2013 until 2021, when it started rising slowly. The later rise is also, however, less in reality if reviewed against the dropping values of the naira due to inflation. This implies that the average annual healthcare budget has not kept pace with population growth or inflation, leading to inadequate funding for essential services. For instance, although the annual budget per individual moved from approximately ₦1,005.55 ($8.22, using average exchange rate for the year) in 2010 to around ₦4,806.85 ($7.57) by 2023, this increase does not reflect the healthcare financial needs in the country since expenditure on healthcare per individual per annum was between ₦9,291.76 ($76.00) in 2010 and ₦33,497.14 ($83.00) in 2021 with the peak value of $106.00 in 2014 followed by a downward trend concealed in naira equivalence by the declining exchange rates. The data suggests that Nigeria has one of the highest out-of-pocket expenditures globally, with an average of 91.47% of total health expenditure coming from other sources and patients' pockets, which is over 63% of health spending. This has been identified as a remarkable burden to the healthcare users by studies such as Onwujekwe et al [67], that is greatly exacerbated by a lack of an effective national health insurance system, which covers only between 5% and 8% of the population [43,44].

### GDP, National budget, and health budget correlations

There was a positive correlation between Nigeria's Gross Domestic Product (GDP) and total health budgets, showing that as the economy grows, there is an increasing allocation of resources to healthcare (Fig 6). However, despite this upward trend, the health budget as a percentage of GDP has not met the recommended benchmark of 5% but has struggled from 0.48% of GDP in 2010 to 0.61% in 2023 (Fig 9). Such persistently low performance highlights a need for higher and sustained investment to align with global standards to improve health outcomes.

Of note also is that the healthcare budget, as a percentage of the total national budget, has experienced a downward trend, with the highest being 5.78% in 2015 and the lowest being 3.46% in 2022. This contrasts with many published and often misleading claims that the Nigerian government budget is as high as 4% to 7% of the national budget for healthcare. Aside from the media outlets, studies such as those of Onwujike et al [67] cited prominent sources making similar statements: *"Sometimes it (government budget for health) is 4% of the total budget, sometimes, it is 6%. I think the best we ever had is about 7%".* In addition to inaccuracies in such claims, the actual values are way below the 2001 Abuja declaration that sets 15% of the national budget as a benchmark for healthcare and way off the actual healthcare spending on the population.

The analysis showed that average health expenditure from 2010 to 2021 was $82.75 per person. Nigerians expended an equivalent of between 3.47% and 5.97% of GDP on healthcare, declining from 5.97% in 2012 to 4.06% in 2017 and 2021, with further drops in between. Although this data is easily accessible, individuals must recognise that it might be misleading for those who do not exercise caution in understanding the substantial differences between *expenditure* and *actual budgets* in this context. The fluctuations in the health expenditure relative to GDP are primarily due to the rising GDP, rather than a decrease in healthcare expenditures, as total health spending has remained relatively stable, exhibiting a steady rise from ₦9,291.76 ($76.00) to ₦33,084.78 ($83.00) in 2023 while the budgeted part of it has steadily dropped from 10.82% in 2010 to as low as 7.20% in 2021.

Consequently, claiming also to allocate about 5% of GDP to health does not inherently indicate that the funding originates from the government budget, as elucidated above and in the preceding sections. The disparity between published and implied statistics can generate misinterpretations that can impede adequate healthcare and increase health disparities among the population who are bearing the cost burden, with slight amelioration from other sources, like donor funds [59].

### Implications of budgetary deficits

Historical data show that despite government claims of efficient budget utilisation, many citizens perceive a disconnect between budget allocations and actual service delivery [12,67]. This aligns with findings from Gyawali et al. [1], who emphasise that effective healthcare financing must be accompanied by transparency and accountability to ensure that funds effectively improve healthcare services. Moreover, international comparisons reveal that Nigeria's healthcare financing lags behind countries like Denmark, South Korea, and Taiwan, which have successfully implemented comprehensive health systems prioritising accessibility and quality [4]. The gaps between healthcare outcomes in Nigeria and these countries also underscore the urgent need for Nigeria to adopt best practices from these nations to enhance its healthcare financing strategies.

### Major challenges identified by users

Respondents identified several critical challenges within the Nigerian healthcare system that hinder access to quality care. The most frequently mentioned issues included inadequate infrastructure, high out-of-pocket costs, insufficient staffing, and poor administrative efficiency. These challenges reflect systemic inefficiencies and resource constraints affecting healthcare delivery in Nigeria [14]. Nigeria has not prioritised the local production of healthcare goods, and therefore consistently depends on imported medical goods and devices to deliver services to the needy population [73]. High out-of-pocket costs are particularly concerning as they exacerbate health inequalities and deter individuals from seeking necessary medical care [37]. Evidence suggests that these barriers significantly impact healthcare access for lower-income populations and contribute to a growing trend of medical tourism among wealthier Nigerians seeking better care abroad [15,16].

Moreover, the administrative inefficiencies within Nigeria's healthcare system have been linked to corruption and mismanagement of resources. As highlighted by Adeloye et al. [14], these issues create an environment where quality care is inaccessible and unaffordable for many citizens. The need for improved governance and accountability within the healthcare sector cannot be overstated.

### Perceived improvements and suggestions for change

While some users acknowledged improvements in specific areas, such as better access to primary healthcare facilities over the past 10 years, these changes were not uniform across the regions. Respondents suggested several key areas for improvement, such as increasing government funding, strengthening health insurance schemes, enhancing infrastructure and staffing levels, and improving administrative processes.

These recommendations reflect a consensus among users on critical areas needing attention to enhance overall healthcare quality [51]. The emphasis on increasing government funding and strengthening health insurance schemes highlights an urgent need for systemic reforms to address users' financial barriers. Furthermore, administrative transparency is crucial for building trust between providers and users. As noted by Yong and Oslen [74], effective governance can significantly enhance user experience by ensuring accountability in service delivery.

International perspectives on successful health systems emphasise integrating user feedback into policy-making processes. For instance, countries like Denmark have implemented robust mechanisms for citizen engagement in health policy discussions, resulting in improved satisfaction rates among users [75]. Adopting similar approaches could help Nigeria align its healthcare services with user needs better.

### Key summary of findings

1. *Low Global Ranking*: Nigeria seeks to improve its healthcare from its ranking of 84th among 110 nations in CEOWORLD Magazine Health Care Index and 146 out of 166 in the Sustainable Development Goals (SDGs) index.

2. *Rising Demand for Services*: There is an increasing demand for improved healthcare services due to a rise in population, disease burden, globalisation, and greater patient awareness, which puts pressure on the healthcare system to achieve adequate funding and care quality.

3. *Quality Gaps*: The quality ratings of healthcare systems: 74.09% for healthcare structure, 61.67% for services, and 60.89% for cost, reflect significant systemic gaps in care delivery.

4. *Healthcare Utilisation*: A significant number (87.36%) of survey participants reported using healthcare facilities at least once a year, highlighting ongoing engagement with the system despite prevalent dissatisfaction, which may suggest a widespread lack of better options.

5. *Financial Burden*: A substantial number (85.00%) of users paid Out-of-Pocket for healthcare services, with 72.60% expressing dissatisfaction with the value received for their expenditures, indicating a heavy financial burden.

6. *Preference for Government Facilities*: Although 71.43% preferred government facilities over private ones, a good portion (28.63%) of respondents would rather use private facilities due to significant challenges in accessing quality care within these institutions.

7. *Identified System Challenges*: Key challenges included high costs (62.75%), poor funding (85.75%), inadequate services (92.76%), and limited staffing (90.73%), all contributing to a negative healthcare experience.

8. *Professional Competence Concerns*: A significant number of respondents reported issues with incompetent healthcare professionals (83.13%) and poor administrative processes (87.80%) as key factors undermining trust in the system.

9. *Access Issues*: Respondents highlighted barriers such as long distances to healthcare facilities (74.10%) and a lack of essential medicines (88.47%), which hinder access to necessary care.

10. *Corruption and Mismanagement*: Corruption and fund misappropriation were major concerns, with 92.36% of respondents citing these as significant issues affecting the effectiveness of the healthcare system.

11. *Budgetary Analysis Findings*: The analysis revealed that government spending on healthcare averaged approximately $7.12 per person annually and has declined over the past decade, from around $10.47 in 2012 to $7.57 in 2023, indicating insufficient investment in health services.

12. *Need for More Robust Healthcare Budgeting:* Nigeria's healthcare funding is critically inadequate, with only an average of 0.37% of its GDP allocated to healthcare. This leads to a heavy reliance on Out-of-Pocket payments and donor contributions for health services.

13. *Need for Policy Reforms:* The study emphasises the urgent need for policy reforms and implementation to improve healthcare financing and quality among users.

## Recommendations

Based on the findings of this study, several strategic recommendations can be made to improve the Nigerian healthcare system:

1. *Increase Budgetary Allocation for Public Health:* The Nigerian government should prioritise increasing its healthcare budget to ensure adequate funding for services and infrastructure improvements.

2. *Expand Health Insurance Coverage:* Comprehensive health insurance schemes can reduce users' out-of-pocket expenses and increase access to care. The NHIS should engage residents at grassroot levels to capture the informal systems for better enrolment and fund pooling.

3. *Create an Efficient Insurance Awareness Program:* Develop an Effective Insurance Awareness Initiative: Establish a promotional program to enhance Nigerians' understanding and appreciation of insurance practices to drive enrolment. The enrolment process can be tied to common processes like National Identification Number (NIN) and Bank Verification Number (BVN) registration, especially for non-government employees.

4. *Invest in Infrastructure Development:* Upgrading existing healthcare facilities and establishing new ones in underserved areas will enhance service delivery.

5. *Enhance Workforce Conditions:* Improving working conditions for healthcare professionals through better remuneration and career development opportunities can help retain skilled workers.

6. *Implement Evaluation Systems:* Establishing robust mechanisms for evaluating healthcare quality based on user feedback can guide policy adjustments and service improvements.

These recommendations are also supported by literature emphasising the importance of systemic reforms in healthcare financing and delivery [51,76].

## Conclusion

Transformative advancements in Nigeria's healthcare financing are urgently needed to bridge the gap between budget allocations and user experiences. Although strides have been made in global rankings over the past decade, local challenges remain deeply entrenched, underscoring the necessity for immediate action. The study reveals widespread dissatisfaction among Nigerian residents. As user satisfaction is a key performance metric of healthcare quality, policymakers must prioritise transparency, accountability, and user feedback in decision-making. Similarly, significant deficiencies in healthcare financing and budgeting necessitate a collaborative approach among government agencies, healthcare providers, and users. Focused policy developments and implementations are required to enhance funding, infrastructure, and stakeholder engagement to create a more equitable and efficient healthcare system that meets the population's needs. Such advancements will improve healthcare outcomes, foster community trust, and contribute to economic growth and stability, as a healthy population is fundamental to national progress.

## Supporting information

**S1 Checklist. S1 Inclusivity in global research Checklist.** S1 Checklist is a PLOS policy on inclusivity in global research, which focuses on ensuring that research conducted outside of a researcher's home country is reported transparently and ethically.
(DOCX)

**S1 Table. Detailed Chi-Squared Crosstabulation.** S1 Table provides a detailed representation of the crosstabulation of Sociodemographic attributes and average rating of the quality of the healthcare system.
(DOCX)

**S2 Table. Detailed Chi-Squared Crosstabulation.** S2 Table provides a detailed representation of the crosstabulation of sociodemographic attributes and preference for healthcare facilities.
(DOCX)

## Author contributions

**Conceptualization:** Blessing O. Josiah, Brontie Albertha Duncan, Lordsfavour Anukam, France Ncube, Oghosa Gabriel Josiah, Ajao Adewale Gbolabo, Olukayode Joseph Oladimeji, Timothy Wale Olaosebikan, Marios Kantaris.

**Data curation:** Blessing O. Josiah, Emmanuel Chukwunwike Enebeli, Brontie Albertha Duncan, Lordsfavour Anukam, Muhammad Baqir Shittu, Mercy Emmanuel, Oyinye Prosper Martins-Ifeanyi, Abosede Peace Adebayo,

Busiroh Mobolape Ibraheem, Ubiebo Ataisi Ekenekot, Mudiaga Sidney Edafiejire, Solomon Oluwaseun Olukoya, Ufuomaoghene Jemima Mukoro, Siyouneh Baghdasarian, Joy Chioma Obialor, Gloria Oluwakorede Alao, Blessing Onyinye Obialor, Ndidi Louis Otoboyor, Oghosa Gabriel Josiah, Precious Ebinehita Imoyera, Blessing Chiamaka Nganwuchu.

**Formal analysis:** Blessing O. Josiah, Emmanuel Chukwunwike Enebeli, Brontie Albertha Duncan, Muhammad Baqir Shittu, Fawole Israel Opeyemi, Joshua Okonkwo, Olukayode Joseph Oladimeji, Marios Kantaris.

**Investigation:** Blessing O. Josiah, Emmanuel Chukwunwike Enebeli, Brontie Albertha Duncan, Lordsfavour Anukam, Kelechi Eric Alimele, Oluwadamilare Akingbade, Abosede Peace Adebayo, Busiroh Mobolape Ibraheem, Ubiebo Ataisi Ekenekot, Mudiaga Sidney Edafiejire, Solomon Oluwaseun Olukoya, Ufuomaoghene Jemima Mukoro, Joy Chioma Obialor, Gloria Oluwakorede Alao, Blessing Onyinye Obialor, Ndidi Louis Otoboyor, Oghosa Gabriel Josiah, Joshua Okonkwo, Precious Ebinehita Imoyera, Blessing Chiamaka Nganwuchu.

**Methodology:** Blessing O. Josiah, Emmanuel Chukwunwike Enebeli, Brontie Albertha Duncan, Prisca Olabisi Adejumo, Muhammad Baqir Shittu, France Ncube, Oluwadamilare Akingbade, Ajao Adewale Gbolabo, Olukayode Joseph Oladimeji, Marios Kantaris.

**Project administration:** Blessing O. Josiah, Emmanuel Chukwunwike Enebeli, Brontie Albertha Duncan, Lordsfavour Anukam, Mercy Emmanuel, Marios Kantaris.

**Resources:** Blessing O. Josiah, Emmanuel Chukwunwike Enebeli, Brontie Albertha Duncan, Chinelo Cleopatra Josiah, Kelechi Eric Alimele, Mercy Emmanuel, Oyinye Prosper Martins-Ifeanyi, Fawole Israel Opeyemi, Oluwadamilare Akingbade, Abosede Peace Adebayo, Busiroh Mobolape Ibraheem, Ubiebo Ataisi Ekenekot, Mudiaga Sidney Edafiejire, Solomon Oluwaseun Olukoya, Ufuomaoghene Jemima Mukoro, Siyouneh Baghdasarian, Joy Chioma Obialor, Gloria Oluwakorede Alao, Blessing Onyinye Obialor, Ndidi Louis Otoboyor, Oghosa Gabriel Josiah, Joshua Okonkwo, Precious Ebinehita Imoyera, Blessing Chiamaka Nganwuchu, Timothy Wale Olaosebikan.

**Software:** Blessing O. Josiah, Chinelo Cleopatra Josiah, Muhammad Baqir Shittu, Joshua Okonkwo.

**Supervision:** Blessing O. Josiah, Emmanuel Chukwunwike Enebeli, Brontie Albertha Duncan, Lordsfavour Anukam, France Ncube, Mercy Emmanuel, Oyinye Prosper Martins-Ifeanyi, Timothy Wale Olaosebikan, Marios Kantaris.

**Validation:** Blessing O. Josiah, Emmanuel Chukwunwike Enebeli, Brontie Albertha Duncan, Chinelo Cleopatra Josiah, Muhammad Baqir Shittu, France Ncube, Kelechi Eric Alimele, Mercy Emmanuel, Fawole Israel Opeyemi, Oluwadamilare Akingbade, Abosede Peace Adebayo, Busiroh Mobolape Ibraheem, Ubiebo Ataisi Ekenekot, Ufuomaoghene Jemima Mukoro, Joy Chioma Obialor, Gloria Oluwakorede Alao, Blessing Onyinye Obialor, Ndidi Louis Otoboyor, Joshua Okonkwo, Precious Ebinehita Imoyera, Ajao Adewale Gbolabo, Blessing Chiamaka Nganwuchu, Timothy Wale Olaosebikan, Marios Kantaris.

**Visualization:** Blessing O. Josiah, Emmanuel Chukwunwike Enebeli, Brontie Albertha Duncan, Chinelo Cleopatra Josiah, Muhammad Baqir Shittu, Kelechi Eric Alimele, Oyinye Prosper Martins-Ifeanyi, Oluwadamilare Akingbade, Busiroh Mobolape Ibraheem, Ubiebo Ataisi Ekenekot, Mudiaga Sidney Edafiejire, Solomon Oluwaseun Olukoya, Ajao Adewale Gbolabo, Marios Kantaris.

**Writing – original draft:** Blessing O. Josiah, Emmanuel Chukwunwike Enebeli, Brontie Albertha Duncan, Prisca Olabisi Adejumo, Chinelo Cleopatra Josiah, Lordsfavour Anukam, Muhammad Baqir Shittu, Mercy Emmanuel, Oyinye Prosper Martins-Ifeanyi, Fawole Israel Opeyemi, Oluwadamilare Akingbade, Siyouneh Baghdasarian, Joshua Okonkwo, Precious Ebinehita Imoyera, Marios Kantaris.

**Writing – review & editing:** Blessing O. Josiah, Emmanuel Chukwunwike Enebeli, Brontie Albertha Duncan, Prisca Olabisi Adejumo, Chinelo Cleopatra Josiah, Muhammad Baqir Shittu, France Ncube, Kelechi Eric Alimele, Fawole

Israel Opeyemi, Oluwadamilare Akingbade, Siyouneh Baghdasarian, Ndidi Louis Otoboyor, Oghosa Gabriel Josiah, Marios Kantaris.

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
