## [Decision Letter · Decision Letter 0]

19 Mar 2025

PGPH-D-24-02908

Critical Review of Healthcare Financing and System Quality Perception among Healthcare Users in Nigeria

Dear Dr. Josiah,

Thank you for submitting your manuscript to PLOS Global Public Health. After careful consideration, we feel that it has merit but does not fully meet PLOS Global Public Health’s publication criteria as it currently stands. Therefore, we invite you to submit a revised version of the manuscript that addresses the points raised during the review process.

We look forward to receiving your revised manuscript.

Kind regards,

Ejemai Eboreime, MD, MSc, PhD

Academic Editor

Journal Requirements:

2. Figure 1: please (a) provide a direct link to the base layer of the map (i.e., the country or region border shape) and ensure this is also included in the figure legend; and (b) provide a link to the terms of use / license information for the base layer image or shapefile. We cannot publish proprietary or copyrighted maps (e.g. Google Maps, Mapquest) and the terms of use for your map base layer must be compatible with our CC-BY 4.0 license.

Additional Editor Comments (if provided):

Reviewers' comments:

Reviewer's Responses to Questions

**Comments to the Author**

1. Does this manuscript meet PLOS Global Public Health’s publication criteria ? Is the manuscript technically sound, and do the data support the conclusions? The manuscript must describe methodologically and ethically rigorous research with conclusions that are appropriately drawn based on the data presented.

Reviewer #1: Yes

Reviewer #2: Yes

2. Has the statistical analysis been performed appropriately and rigorously?

Reviewer #1: Yes

Reviewer #2: Yes

3. Have the authors made all data underlying the findings in their manuscript fully available (please refer to the Data Availability Statement at the start of the manuscript PDF file)?

Reviewer #1: Yes

Reviewer #2: Yes

4. Is the manuscript presented in an intelligible fashion and written in standard English?

Reviewer #1: Yes

Reviewer #2: Yes

5. Review Comments to the Author

Reviewer #1: 1. Verification of Data and Rankings:

The manuscript states that Nigeria ranks 74th out of 89 countries and is 159th out of 162 on the SDG index.

Clarifying the context behind the “89 countries” which was a standalone statistic in the abstract would be helpful.

2. Updating Index Data:

If available, please cite the most recent statistics for the background. For example, the updated SDG report by Sachs et al (2024) ranks Nigeria 146th of 166 countries, rather than the previous 159th position out of 162 countries in 2019.

Incorporating updated data can help strengthen the policy implications of this important study.

3.Healthcare Expenditure/Secondary Data

Please, verify the claim that Nigeria allocates only 0.37% of GDP to healthcare. Provide the source and context (e.g., whether this was based on government data, a questionnaire, or budget analysis from the nine selected states).

Please, clearly explain the methodology used to obtain the budget data before the analyses, clarifying the specific reputable government and online sources in Refs 40 - 42 and whether all the data are publicly available, or were privately sourced.

The transparency of the budget data claims in this study is especially important to support this interesting argument in the discussion: "This contrasts with many published and often misleading claims that Nigerian government budget is as high as 4% to 7% of national budget for healthcare".

4. Additional Clarifications and Methodological Details

In reporting the sample size calculation, please include an appropriate citation (e.g Cochran, 1977).

5. Punctuation and Text Formatting Consistency:

Please, carefully address the typing errors and formatting inconsistencies throughout the manuscript. For example,

check the sentence:

“The survey was conducted between 6/8/2023 and 8/18/2023 and the data was collated with Google…”

and ensure that dates, figures, and numerical data (e.g., population numbers, financial figures) are consistently formatted, and et al is italicized.

Re-examine statements such as “Using (the)Lagos, with the highest adult population”, and a couple more across the manuscript for clarity and proper punctuation.

6. Affiliations and Institutional Details:

Confirm that all author affiliations are complete and accurate. For instance, verify that entries such as “Irrua Specialist Teaching Hospital” include the necessary details (state, country, etc.)

Reviewer #2: I eulogize the authors for the good work done. It was a very good study which highlighted the challenges confronting the healthcare delivery system in Nigeria. I observed some minor issues in the manuscript that the authors need to address. Also, some sentences and paragraphs were repeated which should be deleted. I also observed that some percentages were written to two decimal places while all the others were to just one decimal space, the authors should look into that as well. By and large, it was a well-written manuscript.

6. PLOS authors have the option to publish the peer review history of their article (what does this mean? ). If published, this will include your full peer review and any attached files.

**Do you want your identity to be public for this peer review?** For information about this choice, including consent withdrawal, please see our Privacy Policy .

Reviewer #1: No

Reviewer #2: **Yes: ** Ezra Olatunde Ogundare

---

## [Editor Report · Decision Letter 1]

22 Apr 2025

Critical Review of Healthcare Financing and a Survey of System Quality Perception among Healthcare Users in Nigeria (2010-2023)

PGPH-D-24-02908R1

Dear Josiah,

We are pleased to inform you that your manuscript 'Critical Review of Healthcare Financing and a Survey of System Quality Perception among Healthcare Users in Nigeria (2010-2023)' has been provisionally accepted for publication in PLOS Global Public Health.

Best regards,

Ejemai Eboreime, MD, MSc, PhD

Academic Editor
